

# An analysis coordinate transform to facilitate use of in-situ aircraft observations for flux estimation

Ariana L. Tribby[1], Paul O. Wennberg[2]

[1]Division of Chemistry and Chemical Engineering, California Institute of Technology, Pasadena, California 91125
[2]Division of Engineering and Applied Science, California Institute of Technology, Pasadena, California 91125

*Correspondence to*: Ariana L. Tribby (aaalt02013@gmail.com) or Paul O. Wennberg (wennberg@caltech.edu)

**Abstract.** Analysis of aircraft observations of atmospheric trace gases is key towards improving our understanding of fundamental chemical processes and quantifying anthropogenic emissions. A common approach for such analysis is use of chemical transport models to produce 4-D fields for comparison with these observations together with various inversion
techniques to constrain the underlying fluxes and chemistry. Yet, time and monetary constraints of expensive computational jobs for chemical transport modelling can be a significant hindrance. Here, we show the advantages of using potential temperature as a dynamical coordinate to compare such simulations to aircraft observations of trace gases whose concentration fields are strongly influenced by synoptic-scale transport. We use global observations of ethane and propane from the Atmospheric Tomography (ATom) aircraft mission and simulate global mole fractions for these gases using GEOS-
Chem High Performance v13.4.1. We show, using potential temperature as an analysis coordinate, that Bayesian estimates of the fluxes of these gases in the Northern Hemisphere are largely invariant ($\pm$ 10%) even as the simulation spatial resolution is increased 100-fold. Our approach can have broad applications for the modelling of trace gases in the extra-tropics, particularly those with lifetimes long compared to synoptic timescales.

## 1. Introduction

A common approach for estimating fluxes of trace gases to the atmosphere involve comparing atmospheric simulations with in-situ observations. Such studies enable the use of sparse observations made by aircraft, for example (Xiao et al., 2008; Zhou et al., 2019; Frankenberg et al., 2016). However, computational load can limit the feasibility of these comparisons. Often, efforts are focused on increasing the temporal and spatial resolution of the simulations as a means of capturing the fine scale structure that adds variance to the 4-D chemical fields. This significantly increases the computational load and
analysis time.

Motivated by previous analyses of column $CO_2$ observations (Keppel-Aleks et al., 2011, 2012), we illustrate how potential temperature ($\theta$, in units of Kelvin), can be useful in model-observation comparisons, particularly when the trace gases under analysis have fields that are largely controlled by synoptic-scale dynamics acting on zonal gradients. Variability in the extra-

tropics induced by large-scale circulation patterns can be well-captured since $\theta$ is conserved following adiabatic flow. Trace gases that have longer lifetimes than the typical duration of synoptic-scale meteorology (around 10 days, as described in Wohltmann, (2005)), will generally be well correlated with $\theta$. (The correlation does not hold in the tropics or boundary layer, or when the photochemical lifetimes are short, i.e., during the summer months.)

In this study, we use global aircraft observations of ethane ($C_2H_6$) and propane ($C_3H_8$) from Atmospheric Tomography (ATom) mission during 2016-2017 and simulate those global fields using the chemical transport model, GEOS-Chem High Performance v13.4.1. We show that when used as a zonal coordinate, $\theta$ improves model/observation correlation compared with using classical latitude, longitude, altitude and time coordinates. We use the Bayesian hierarchical model from Tribby et al., (2022) to evaluate flux estimates from simulations performed at 4° x 5°, 2° x 2.5°, and 0.5° x 0.625° resolutions, and find

that when using $\theta$, the comparisons have negligible difference. As such, $\theta$ offers an opportunity to optimize time and cost of model simulations for certain trace gases, several of which have important climate implications.

## 2. Methods

### 2.1 ATom observations

The Atmospheric Tomography (ATom) aircraft campaign comprised of four sequential global flights during 2016-2018. We

use two flight campaigns during July-August 2016 and January-February 2017.

We exclude from our analysis stratospheric observations (using $N_2O$) and data influenced by highly local emission sources, i.e., boundary layer and biomass burning (Tribby et al., 2022). When using $\theta$ as a vertical interpolation coordinate, we exclude the summer months from our analysis, since the lifetime of $C_2H_6$ and $C_3H_8$ are reduced, and regional/local sources

dominate the variance. Further, we exclude subtropical transport by limiting observations with tropopause pressure above 100 hPa (only about 5% of data was affected by this constraint), which sufficiently reduced subtropical influence.

We show flight paths for the data above 20 degrees north in Figure 1, as we only consider northern fluxes since most emissions of short-lived $C_2H_6$ and $C_3H_8$ originate in the northern hemisphere, and their lifetimes are shorter than norther-

southern hemispheric exchange rate.



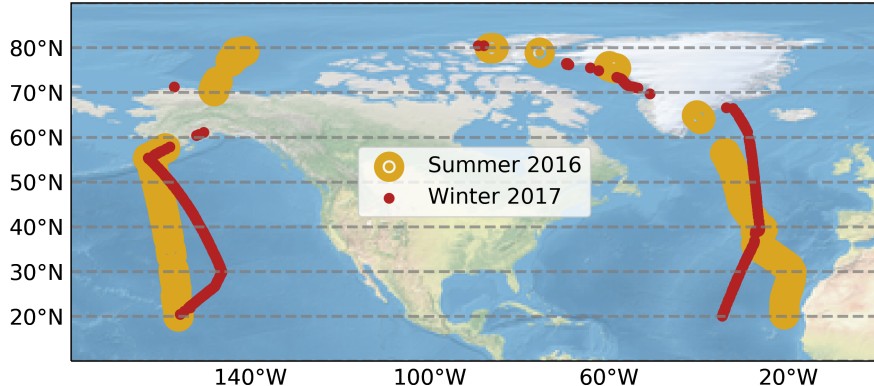

**Figure 1. Flight paths during ATom flight campaign. We use summer 2016 and winter 2017 campaigns over the Atlantic Ocean in our analysis. We only consider observations above 20 degrees north, as explained in the methods. We use Cartopy (Elson et al., 2018) to generate the base map in this figure.**

## 2.2 GEOS-Chem simulations

We simulated ATom aircraft observations using the GEOS-Chem classic global 3-D chemical transport model in v13.4.1 (The International GEOS-Chem User Community, 2022) on Amazon Web Services (AWS) using a public GEOS-Chem Amazon Machine Image (ami-0491da4eeba0fe986; (Zhuang et al., 2019)). We used the standard full chemistry option and 3 horizontal resolutions to simulate ATom 1 and 2, including 4° x 5°, 2° x 2.5°, and 0.5° x 0.625°, all with the native 72 hybrid sigma/pressure levels using MERRA-2 reanalysis meteorology products by the Global Modeling and Assimilation Office (GMAO) at NASA Goddard Space Flight Center (Gelaro et al., 2017), available on the AMI. For all 3 horizontal resolution simulations, we output hourly simulations over the ATom 1 and 2 flight campaign periods.

We conducted a global simulation for 4° x 5° and 2° x 2.5° horizontal resolutions before sampling to the aircraft path. Default chemistry and configurations were used for 4° x 5° and 2° x 2.5°, except for custom $C_2H_6$ and $C_3H_8$ emissions, described below. We used a 1-year spin-up at 4° x 5° horizontal resolution, followed by a 10-day spin-up with the 2° x 2.5° or 0.5° x 0.625° meteorological files.

The 0.5° x 0.625° simulations were conducted using a nested grid over a custom box that encompassed the ATom Atlantic curtain transect. The vertices of our custom grid included a minimum/maximum latitude of (18.0, 88.0) degrees north and a minimum/maximum longitude of (-90.0, -15.0) degrees west with a 3-grid buffer for all sides. We generated global boundary condition tri-hourly daily files for our custom grid using a 4° x 5° simulation and custom emissions for $C_2H_6$ and $C_3H_8$,

described below. During the nested run, several species were flagged as having negative values during PBL mixing in GEOS-Chem, including $HNO_3$, $NH_3$, $NO$, $NO_2$, $O_3$, and halogen chemistry sea salt alkalinity variables, "SALAAL" and "SALCAL", causing the simulations to end. Manually increasing the background concentrations or reducing the transportation/convection timestep to 150 seconds and the chemistry/emission timestep to 300 seconds did not prevent these issues. We kept the timesteps at those reduced values and edited the wrapper module, "mixing_mod.F90", to replace negative values with zero to allow the model to run. No other species were flagged as having negative values.

Emissions for $C_2H_6$ and $C_3H_8$ were computed using the Harmonized Emissions Component (HEMCO) Standalone version 3.5.0-rc.1 (Yantosca et al., 2022b) and GEOS-Chem 14.0.0-rc.1 (Yantosca et al., 2022a) on AWS using a public Amazon Machine Image (ami-0491da4eeba0fe986). We revised the default emissions using the same methods from Tribby et al., (2022): we scaled all sectors of default $C_2H_6$ by 1.1 and we substituted $C_3H_8$ with default $C_2H_6$ before scaling by 1.2. Software packages used in this analysis include Matplotlib (Caswell et al., 2022; Hunter, 2007) and Pandas (Reback et al., 2021).

## 3. Results

In Figure 2, we show transect "curtain" plots of $C_3H_8$ for 4° x 5°, 2° x 2.5°, and 0.5° x 0.625° horizontal resolution GEOS-Chem simulations. ($C_2H_6$ is shown in Figure S2.) We interpolate 4° x 5°, 2° x 2.5° along pressure and latitude to the 0.5° x 0.625° scale using nearest-neighbour techniques. As expected, 0.5° x 0.625° show more defined structures. However, if we instead create the same curtain plots but using $\theta$ as the vertical coordinate (Figure 3), the three simulations look very similar. This holds true for the winter 2017 observations, but not for the summer 2016 observations, since the increased sunlight and resulting increase in oxidation rate reduces the chemical lifetime of $C_2H_6$ and $C_3H_8$, resulting in a poor relationship with $\theta$ (Figures S3 and 4).







**Figure 2. GEOS-Chem-simulated C₃H₈ (ppb) Atlantic Ocean transect along pressure and latitude. Simulations were conducted for the ATom winter 2017 campaign time period. A representative simulation transect was selected by slicing along a longitude of -30.0 degrees W and a single time point during the day shown in the column label. (We used the average longitude encountered along the Atlantic flight track. Latitude was not interpolated to exactly match the aircraft flight path in this figure.) Horizontal resolutions of 4 x 5 and 2 x 2.5 were interpolated along latitude and pressure to 0.5 x 0.625 resolution. The bottom panel is a**



zoomed-in illustration of row: 3, column: 2, with the aircraft flight path represented by the grey line, the aircraft observations shown in triangle markers, and potential temperature contours shown in black lines. C₂H₆ is included in the SI.

**Figure 3. GEOS-Chem-simulated C₃H₈ (ppb) Atlantic Ocean transect along potential temperature and latitude. Simulations were conducted for the ATom winter 2017 campaign time period. A representative simulation transect was selected by slicing along a**



**longitude of -30.0 degrees W and a single time point during the day shown in the column label. (We used the average longitude encountered along the Atlantic flight track. Latitude was not interpolated to exactly match the aircraft flight path in this figure.) Horizontal resolutions of 4 x 5 and 2 x 2.5 were interpolated along potential temperature and latitude to 0.5 x 0.625 resolution. The bottom panel is a zoomed-in illustration of row: 3, column: 2, with the aircraft flight path represented by the grey line and the aircraft observations shown in triangle markers. $C_2H_6$ is included in the SI.**

When we interpolate the GEOS-Chem simulations along aircraft flight path latitude, longitude, time, and $\theta$, we see good agreement between simulations and the aircraft observations, consistent across all 3 horizontal resolutions (Figure 4). Furthermore, there is generally good agreement between the aircraft observations and simulations 5 days before and after the flight path, which is expected during winter months when $C_3H_8$ and $C_2H_6$ have longer lifetimes and therefore higher atmospheric abundance in the Northern Hemisphere.

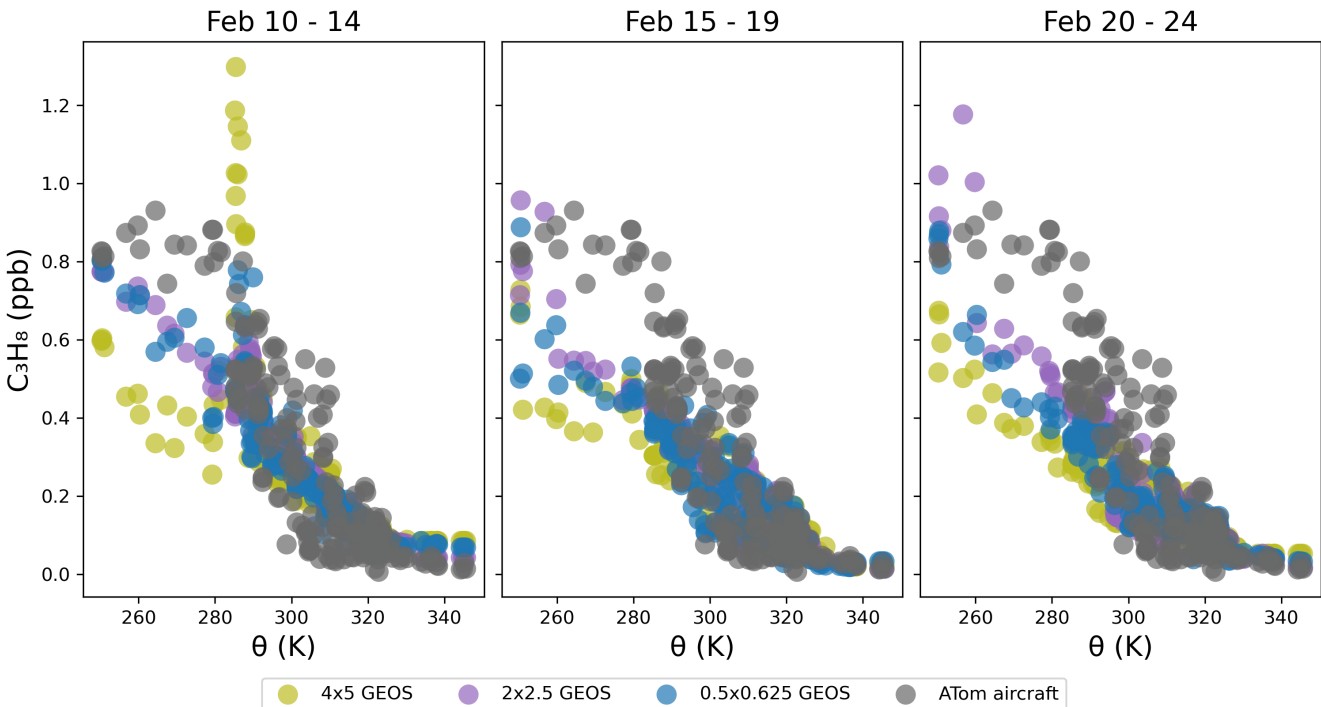

**Figure 4. $C_3H_8$ vs potential temperature. GEOS-Chem simulations were conducted for the ATom winter 2017 campaign time-period and were interpolated along aircraft flight path latitude, longitude, time, and potential temperature. Feb. 15-19[th] is the observed aircraft flight path time, and the left and right plots show times not actually observed by the aircraft, and instead are GEOS-Chem simulations sampled 5 days before/after the plane start time. $C_2H_6$ included in the SI.**



We compare the aircraft observations with the GEOS-Chem simulations using the same Bayesian hierarchical model from Tribby et al., (2022) to capture the contribution of uncertainty due to transport in GEOS-Chem. In summary, Tribby et al., (2022) assumed differences between the GEOS-Chem simulations and aircraft observations are largely dependent on the
underlying emissions grid during the winter when there is decreased sunlight/oxidation, such that

$$a = gcs \cdot \alpha, \tag{1}$$

where $a$ is the aircraft mole fraction, $gcs$ is the GEOS-Chem simulation, and $\alpha$ is a scalar that quantifies the difference
between the simulations and the aircraft observations, which directly attributes the missing emissions. We expect $\alpha$ to be close to 1, since in our simulations here, we updated the emission fluxes to those estimated in Tribby et al., (2022) before running the GEOS-Chem simulations. We follow all methods of the previous study, including sampling the GEOS-Chem simulations several days before and after the aircraft flight latitude, longitude, and time before interpolating on the vertical level using $\theta$. Please refer to Tribby et al., (2022) for the complete statistical model and its derivation, the software used, and
the development of the priors.

With $\theta$ as the interpolation coordinate, the Bayesian results are similar regardless of the simulation grid scale (Table 1): the mean posterior varies 10% or less between all horizontal resolutions for both $C_3H_8$ and $C_2H_6$. The relative spread of the credible interval was about the same for the finest and coarsest resolutions. As a test, we substituted the 0.5° x 0.625°
simulations as the "observed data" in the Bayesian model, with the 4° x 5° as the "simulated data". This resulted in a scalar whose 95% credible interval ([0.71, 0.86] for $C_3H_8$) overlapped with all previous parameters in Table 1. This Bayesian exercise confirms the value of using $\theta$ as the analysis coordinate to decrease time and monetary costs associated with chemical transport modelling.

**Table 1. Bayesian inference results quantifying missing emissions for GEOS-Chem v13.4.1. The 95% credible interval is shown.**

| Spatial Resolution | $\alpha$ Parameter | |
|---|---|---|
| | $C_3H_8$ | $C_2H_6$ |
| 4 x 5 | [0.80, 1.0] | [0.88, 1.1] |
| 2 x 2.5 | [0.81, 1.0] | [0.91, 1.1] |
| 0.5 x 0.625 | [0.90, 1.1] | [0.95, 1.2] |



**Author contributions**

A.L.T performed research and wrote the manuscript. P.O.W. contributed to the interpretation of the results and writing of the manuscript.

**Competing interests**

The authors declare no competing financial interest.

**Acknowledgements**

Part of this work was supported by the Resnick Sustainability Institute, including computations conducted in the Resnick High Performance Computing Center.

**Financial support**

A.L.T. received funding from NASA Grant Award No. 22-SMDSS-0009.

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
