# Peer review of "An analysis coordinate transform to facilitate use of in-situ aircraft observations for flux estimation"

_EGUsphere, 2023_

## Author Comment (AC1)

**Point-by-Point Responses to Anonymous Referee #1**

We thank Anonymous Referee #1 for their time and effort in reviewing our work and appreciate their recommendations for ways to improve the manuscript. Following many of their suggestions, we believe the revised manuscript is an improvement upon the original submission. The referee comments below are taken from https://doi.org/10.5194/egusphere-2023-2227-RC1.

This document provides a point-by-point response to Referee #1. The responses follow the sequence: (1) comments from the referee, (2) authors' response, (3) authors' changes in manuscript.

**1.1 Referee Comment**

"This paper presents a case study of using a previously developed Bayesian approach to evaluate emission estimates of C3H8 and C2H6 from global model simulations and aircraft observations from the ATom campaign. The paper compares 3 different model simulations and demonstrates that the results are less dependent on model resolution when using potential temperature (Tpot) rather than pressure as coordinate. The main conclusion of this study is that using Tpot can safe [sic] efforts and costs since coarse resolution simulations are sufficient and provide similar results as more costly high resolution runs."

**1.2 Author Response**

We substantially revised the manuscript to focus on an analysis approach by creating pseudo data for $C_2H_6$ and $C_3H_8$ as sampled by the DC8 aircraft during the Atmospheric Tomography (ATom) aircraft mission. We sample simulations from the highest resolution (0.5 x 0.625 degree) GEOS-Chem High Performance v14.3.1 and then evaluate the correlation of these pseudo data with lower resolution simulations and simulations sampled +/- 5 days from the pseudo data. We apply a simple statistical analysis that illustrates the value of potential temperature as a vertical coordinate in comparing sparse observations with a GCM.

**1.3 Manuscript Changes**

Figures 4-8 and corresponding SI.

**2.1 Referee Comment**

"The case study per se could be of interest, specifically to other studies using ATom data, modelers or developers of emission inventories, but in my view there is significantly more work needed before the paper is ready for publication and can provide value to the scientific community. Below some of my major concerns:

Throughout the paper the paper lists a number of limitations on when the methodology can be used and they also applied significant filtering to the data set. More information is needed on how the authors decided on the different filtering and how this could be generalized and be applicable to other cases. How could you decide if this method is applicable and valid when multiple resolution simulations are not done?  Have you tested in with other species of lifetimes ~10 days or longer (e.g. CO?)"

**2.2 Author Response**

The referee highlighted the need for providing more information on data filtering techniques to help make the study generalizable and applicable to other cases. Below we provide our response and in section 2.3 we detail where we made changes to the manuscript.

As potential temperature ($\theta$) is conserved following adiabatic flow, it is best used within certain dynamical conditions. The following conditions may limit its use as an effective zonal coordinate:
- $\theta$ is not conserved within moist convection and turbulent conditions, e.g., within the boundary layer.
- Synoptic-scale meteorology has a timescale of about 10 days and a horizontal length scale of greater than 1000 kilometers.[1]

Generally, $\theta$ will provide a more precise coordinate framework when: 1) the trace gasses of interest have longer atmospheric lifetimes than the synoptic meteorology timescales (~2 weeks); and 2) the region of study include conditions where $\theta$ is conserved, e.g., free extra-tropical troposphere and stratosphere.

Consistent with these constraints, our analysis is restricted to the free troposphere and the extratropics. In this region, variability within large-scale circulation can be well-captured using $\theta$ as a zonal coordinate.

Even though $C_3H_8$ and $C_2H_6$ have photochemical  lifetimes longer than typical vertical and horizontal transport during the winter, they are still somewhat sensitive to the a priori zonal distribution of their emissions. We apply a number of constraints to study well-mixed parcels that are independent from any filtering techniques applied to use $\theta$ as a zonal coordinate described above. Thus, we exclude data where highly localized sources influence the mole fraction of these alkanes:
* * *
[1] Jacob, D. J. *Introduction to Atmospheric Chemistry*; Princeton University Press: Princeton, N.J, 1999; pp 52-53.

- We reduce the influence from local plumes by analyzing observations in the free troposphere and over the ocean while excluding observations taken within the boundary layer and over land masses where highly local sources exist such as energy infrastructure.
- Nearby biomass burning emissions are identified and excluded using co-measurements of HCN and CO.
- Summer observations are excluded from the analysis, as high temperatures and OH shortens the lifetime of $C_3H_8$ and $C_2H_6$ such that nearby sources dominate the variance.
- Similarly, observations in the subtropics are sensitive to transport from the extratropics, where most emissions of $C_3H_8$ and $C_2H_6$ originate. We exclude subtropical air parcels using co-measurements with tropopause pressure above 100 hPa (about 5% of data were excluded under this constraint).

Conversely, we exclude alkane observations that are poorly connected to underlying fluxes. Sources of $C_3H_8$ and $C_2H_6$ largely originate from northern hemispheric land masses. The lifetime of $C_3H_8$ and $C_2H_6$ is less than or equal to a few months during the summer, but air parcel mixing between the northern to southern hemispheres is on the order of a year.[2] As a result, the mole fraction of these gasses is relatively low in the southern hemisphere. The relatively short lifetime of $C_3H_8$ and $C_2H_6$ compared to vertical transport owes to the low abundance of these alkanes in the stratosphere. As such, we make the following restrictions:
- We arbitrarily restrict observations above 20 degrees north.
- We exclude stratospheric observations using $N_2O$ as a tracer, which is inert and generally well-mixed in the troposphere but is quickly destroyed in the stratosphere by photolysis and reaction with $O^1D$.[3]

**2.3 Manuscript Changes**

Additional information on these filtering techniques was included in Section 2 Methods (lines 53-138), with an additional section added: "Considerations for θ as a zonal coordinate" to highlight techniques specific to this zonal coordinate. We also added two figures in the SI to show filtering. We added additional information related to generalizability in the Introduction (lines 24-51).

**3.1 Referee Comment**

"The highest resolution tested is 0.5x0.625 deg which is still fairly coarse. I suggest to make very clear that this methodology has only been tested on model resolutions global climate models are generally run at."
* * *
[2] Jacob, D. J. *Introduction to Atmospheric Chemistry*; Princeton University Press: Princeton, N.J, 1999; pp 52-53.
[3] Seinfeld, J. H.; Pandis, S. N. *Atmospheric Chemistry and Physics: From Air Pollution to Climate Change*, Third edition.; John Wiley & Sons: Hoboken, New Jersey, 2016; pp 129.

**3.2 Author Response**

We added this clarifying language throughout the document.

**3.3 Manuscript Changes**

Line 10, 12, 34, 37, 41, 42, 44, 245.

**4.1 Referee Comment**

"Could you please confirm that all simulations use the same base emissions and also state what inventory they are based on. It was not clear from the description. Also in Line 91 it is not clear what is scaled how? You substituted C3H8 with default C2H6? Are there no C3H8 emissions available?"

**4.2 Author Response**

Referee suggests clarifying base emissions used, what inventory they are based on, and how they were scaled.

Emissions for $C_2H_6$ and $C_3H_8$ were computed using a modified version of the Harmonized Emissions Component (HEMCO) Standalone version 3.5.0-rc.1 (Yantosca et al., 2022b) and GEOS-Chem 14.0.0-rc.1 (Yantosca et al., 2022a) on AWS using a public Amazon Machine Image. (Relevant default anthropogenic emissions include Tzompa-Sosa et al. 2017 for $C_2H_6$ and Xiao et al. 2008 for $C_3H_8$.) We revised the default emissions using the same methods from Tribby et al., (2022), as these default inventories were shown to underestimate observed $C_3H_8/C_2H_6$: Briefly, Tribby et al., (2022) they showed that substituting $C_2H_6$ anthropogenic categories for $C_3H_8$ greatly improved resulting simulations of in situ observations. The study additionally quantified observed missing high latitude emissions of these alkanes using Bayesian inference. The resulting revised emissions in Tribby et al. 2022 had good agreement with other studies (Figure 5 and Figure S62). We apply the same methods here by scaling all sectors of default $C_2H_6$ by 1.1 substituting $C_3H_8$ with default $C_2H_6$ before scaling by 1.2. Software packages used in this analysis include Matplotlib (Caswell et al., 2022; Hunter, 2007) and Pandas (Reback et al., 2021).

**4.3 Manuscript Changes**

Lines 111-120 were edited to clarify these methods.

**5.1 Referee Comment**

"It is highly concerning that the simulations experiences negative concentrations and that this has been fixed by simply setting these to zero. Negative concentrations indicate an issue in the model

or the setup and this simple non-physical fix does not provide high confidence in the model results. If this is a general issue and solution with the model and well documented that this does not lead to issues in the simulated fields, then this needs to be referred to in the paper."

**5.2 Author Response**

As explained in the initial manuscript, the $C_3H_8$ and $C_2H_6$ simulated fields did not display negative concentrations. Several aerosol species with no relationship to $C_3H_8$ and $C_2H_6$ did result in negative concentrations, an issue experienced by other users. Fortunately, a new version of GEOS-Chem, v14.3.1, provided a fix for this bug. We repeated the 0.5x0.625 nested simulations with v14.3.1 while still using the same emissions as the 4x5 and 2x2.5 simulations and did not experience these issues with the aerosol species. (Note, only the nested 0.5x0.625 had resulted in negative aerosol.) We include these new simulations for all parts of the analysis. As expected, $C_2H_6$ and $C_3H_8$ changed minimally. Below, we list Figures that incorporate these new simulated fields.

**5.3 Manuscript Changes**

Figure 2-8 and corresponding SI figures.

**6.1 Referee Comment**

" I do not see convincing information that the emission estimates are less resolution dependent using Tpot compared to pressure. E.g., how would Figure 4 look were you to use pressure. Or how would the numbers in Table 1 change if the emission estimates are using pressure as coordinate?"

**6.2. Author Response**

We no longer use the Bayesian approach in this study. Replicating our Bayesian model analysis using pressure proved challenging, as our Bayesian statistical model assumed a good linear correlation between the gas and the vertical coordinate. The correlation of the alkanes with pressure tended to be more scattered. This introduced high uncertainty in our interpretation of the Bayesian results and when comparing the two coordinates against each other. Additionally, we realized that our application of this Bayesian approach was likely confusing to readers since our simulations already incorporated revised emissions and thus we were not expecting the inversion to indicate a large missing source. We anticipated this confusion would be likely to detract from the key point of the manuscript.

Instead, we have substantially revised the manuscript to focus on an analysis approach by creating pseudo data for ethane and propane as sampled by the DC8 aircraft during the Atmospheric Tomography (ATom) aircraft mission. We sample simulations from the highest resolution (0.5 x 0.625 degree) GEOS-Chem High Performance v14.3.1 and then evaluate the

correlation of these pseudo data with lower resolution simulations and simulations sampled +/- 5 days from the pseudo data. We apply a simple statistical analysis that illustrates the value of potential temperature as a vertical coordinate in comparing sparse observations with a GCM.

**6.3 Manuscript Changes**

Figure 4-8.

**7.1 Referee Comment**

"Figure 2 and 3: the 0.5x 0.625 degree runs look significantly different from the coarser resolution results for 23 Feb for both pressure and Tpot. How do the authors explain this?"

**7.2 Author Response**

While some days are predicted to have slightly higher mole fractions, these data did not appear to significantly affect or skew RMS, RMSE, and slope comparisons.

**7.3 Manuscript Changes**

Figures 4-8.

**8.1 Referee Comment**

"Need to specify r2, rmse etc. for Figure 4 and related Figures in Supplement."

**8.2 Author Response**

We have now included RMS, RMSE, and Pearson Correlation Coefficient for this analysis.

**8.3 Manuscript Changes**

Figures 4-8 and related SI figures.

**9.1 Referee Comment**

"The results section needs to be separated into a methodology and an actual results/discussion section."

**9.2 Author Response**

We have moved methodology-related text into the methods section.

**9.3 Manuscript Changes**

Lines 53-138.

**10.1 Referee Comment**

"Methodology: Please be clear how you averaged and filtered the data. To what degree does the +/- 5 days sampling before and after the observation time contribute to reducing the resolution dependence? Effectively you degrade the higher model resolutions more than the coarsest."

**10.2 Author Response**

The referee brings up an interesting point of how the +/- 5 day sampling method may degrade the higher resolution more than the coarser simulations. If this +/- 5 day sampling approach were to have the effect of significantly degrading the higher resolution simulations compared to the coarser ones, we would expect to see a dependence of the strength and uncertainty of the correlation with the sample day, but that is not the case. In Figure 5 the RMSE does not show a clear trend with sample day, and in Figure 8 the daily slope and pearson correlation coefficient do not show a significant reduction in the confidence interval for the actual flight path day, nor does the magnitude of the pearson correlation coefficient significantly vary according to sample day.

**10.3 Manuscript Changes**

We added some discussion on this concept, lines 240-246.

**11.1 Referee Comment**

"Significantly more in-depth analysis and discussion on the results are needed (see comments above). The confidence range in Table 1 does not provide a strong indication for missing sources. Without knowing what the base emissions are, whether they are representative for the year and how they compare to other inventories, you cannot say that there are missing sources but differences could also be due to uncertainties in emission factors, underestimation of emissions from specific sectors etc. It is also not clear to me, if the emissions were scaled upfront by 1.1 and 1.2 for $C_2H_6$ and $C_3H_8$, respectively and the scaling factors (Table 1) are in the order 0.8-1(1.1) and 0.9-1.1(1.2), then the original emissions might not be too low at all if the lower ranges apply."

**11.2 Author Response**

Please see response in section 6.2.

**11.3 Manuscript Changes**

Please see response in section 6.3.

**12.1 Referee Comment**

"Table 1 results for the highest resolution run are actually more different from the two other simulations. This might be related to my question related to Figures 2&3 or simply indicates that there is still some resolution dependence? Have you rerun your simulations with the lower and upper ranges of your estimates to see whether you improve the comparison to aircraft data?"

**12.2 Author Response**

While we have not rerun the simulations with the lower and upper ranges of the revised emissions, RMS and RMSE do not indicate a strong resolution dependence.

**12.3 Manuscript Changes**

Figure 4 and 5.

**13.1 Referee Comment**

"It also needs to be clearly stated how much new information this study provides beyond what has been done in Tribby et al. (2022)."

**13.2 Author Response**

We added clarifying language.

**13.3 Manuscript Changes**

Line 34-36.

**14.1 Referee Comment**
"The paper is missing a Summary/Conclusion."

**14.2 Author Response**
We added this section.

**14.3 Manuscript Changes**
Lines 240-251.

---

## Author Comment (AC2)

**Point-by-Point Responses to Anonymous Referee #2**

We thank Anonymous Referee #2 for their time and effort in reviewing our work and appreciate their recommendations for ways to improve the manuscript. Following many of their suggestions, we believe the revised manuscript is an improvement upon the original submission. The referee comments below are taken from https://doi.org/10.5194/egusphere-2023-2227-RC2.

This document provides a point-by-point response to Referee #2. The responses follow the sequence: (1) comments from the referee, (2) authors' response, (3) authors' changes in manuscript.

**1.1 Referee Comment**

"This paper presents a Bayesian inversion estimating ethane and propane emissions using aircraft data from the ATom campaigns and GEOS-Chem predictions at different horizontal resolutions (4x5, 2x2.5, and 0.5x0.625 degrees nested). The authors argue that the use of potential temperature as a zonal coordinate results in a comparison to aircraft data that is less sensitive to model resolution than comparisons on lat-lon-pressure coordinates, and inversions for longer-lived trace gas species can thus be performed at coarser resolution to save time and cost."

**1.2 Author Response**

We no longer use the Bayesian approach in this study. We substantially revised the manuscript to focus on an analysis approach by creating pseudo data for ethane and propane as sampled by the DC8 aircraft during the Atmospheric Tomography (ATom) aircraft mission. We sample simulations from the highest resolution (0.5 x 0.625 degree) GEOS-Chem High Performance v14.3.1 and then evaluate the correlation of these pseudo data with lower resolution simulations and simulations sampled +/- 5 days from the pseudo data. We apply a simple statistical analysis that illustrates the value of potential temperature as a vertical coordinate in comparing sparse observations with a GCM.

**1.3 Manuscript Changes**

Figures 4-8.

**2.1 Referee Comment**

"While the results of this study certainly have potential value for other work quantifying sources of longer-lived trace gases, this paper is not suitable for publication without major revisions. As is, it assumes the reader is intimately familiar with the previous study on which it is based

(Tribby et al., 2022), and reads as a sort of proof-of-concept of that specific work (a Bayesian inversion based on 4x5 degree model predictions and comparisons to ATom data on potential temperature coordinates). More detail is needed throughout the paper in order for this to serve as a standalone study. A discussion and conclusions section are also needed after the Results section to demonstrate the broader context of this work and its applicability to other research problems (and to complete the paper!). Also, many of the claims made throughout the paper are currently too qualitative, and the authors must do more to quantify their findings and demonstrate how they improve upon inversions using lat-lon-pressure coordinates. I also question whether the analysis used is the best approach for testing their hypothesis. I have made specific line-by-line suggestions below."

**2.2 Author Response**

We have made major revisions to address these concerns. First, we have provided a new analysis approach to test this hypothesis and no longer use the Bayesian approach, we have added a discussion and conclusion section, and more quantitative detail with RMS, RMSE, and simple linear regression analysis.

**2.3 Manuscript Changes**

Figure 4-8.

**3.1 Referee Comment**

"Lines 46-51: More information about the data exclusion process here is needed. Presumably these are being done to limit the dataset to those observations whose variance is dominated by synoptic-scale transport from the midlatitudes? How do the authors confirm their screening can "sufficiently reduce subtropical influence"? How do the results vary if these screening criteria are changed, and what recommendations can the authors make for other studies using this approach?"

**3.2 Author Response**

More information about the data exclusion process was provided, for considerations when using potential temperature as a coordinate (to help generalize this work) as well as filtering specific to the alkanes of interest. We also provide additional figures in the SI related to the filtering methods.

**3.2 Manuscript Changes**

Lines 53-138.

**4.1 Referee Comment**

"Lines 90-93: Please note which emission inventories were used rather than just referring to them as "default". Also, please explain how the scaling factors for the ethane and propane emissions were derived. I think I found the details in Tribby et al. (2022)—they are related to observed ethane:propane ratios?-- but more information is needed here."

**4.2 Author Response**

We added clarifying language for the emissions and addressed a similar question in R1 comments, 4.2 author response.

**4.3 Manuscript Changes**

Lines 111-120.

**5.1 Referee Comment**

"Line 99: While I see how some of variability across latitude is maybe collapsed when moving from pressure to potential temperature coordinates, the differences between Figure 2 and 3 are pretty subtle and I disagree with the authors' claim that the curtain plots in Figure 3 look "very similar" at the different horizontal resolutions. A more quantitative comparison of the differences between simulations is warranted here."

**5.2 Author Response**

We provide quantitative comparisons with our new analysis approach including simple linear regression, RMS, and RMSE.

**5.3 Manuscript Changes**

Figures 4-8.

**6.1 Referee Comment**

"Line 122-127 and Figure 4: A visual inspection is not sufficient here. Please provide some quantitative statistics to demonstrate good agreement between the simulations and aircraft data at all three horizontal resolutions and with the inclusions of the +/- 5 day window when using potential temperature as a coordinate. Also, please show how these statistics are improved relative to a comparison using lat-lon-pressure coordinates. Lastly, why was the 5-day time window chosen for comparison?"

**6.2 Author Response**

We have provided new quantitative comparisons between simulations and observations for pressure and potential temperature as a vertical coordinate and how the statistics were improved

(RMS, RMSE, simple linear regression comparing slope and Pearson Correlation Coefficient). The 5-day time window was chosen arbitrarily to approximate global chemical transport error.

**6.3 Manuscript Changes**

Figures 4-8.

**7.1 Referee Comment**

" Line 137-150: This should all be moved to the Methods section rather than the Results."

**7.2. Author Response**

These methods were removed, as we no longer utilize the Bayesian approach (see RC1, 6.2 author response).

**7.3 Manuscript Changes**

Lines 137-150 (Submission 1) removed.

**8.1 Referee Comment**

"Lines 151-158 and Table 1: This section needs the most additional work to be useful to the broader community. At the very least, it must include a comparison to inversions using lat-lon-pressure coordinates to prove that the resulting alpha is less dependent on model resolution when using potential temperature. Also, while the authors mention they did one test using the 4x5 degree simulated data as the observations in a 0.5x0.625 degree inversion, I question why the authors rely mostly on inversions using real observations rather than simulated ones in this analysis. It seems much easier to quantify the reliability of the results in the latter case, when the true flux is known and can be perturbed. Also, on that note, the authors need to address why the 95% credible interval is [0.71, 0.86] in the simulated inversion—shouldn't it be approaching 1.0 in this case? I'm unclear if that's the case for the inversion using real observations, but more discussion is needed to explain the behaviors here. Also, how dependent are the results on the spatial distribution of the emissions (i.e. if the authors used a prior with the same global flux but different spatial distribution)? Such a test would help demonstrate the limitations and applicability of this work in other contexts."

**8.2 Author Response**

We are no longer utilizing the Bayesian approach (see RC1, 6.2 author response).

**8.3 Manuscript Changes**

Figures 4-8 (new analysis), removal of Table 1.

**9.1 Referee Comment**

"Line 102: I think this should be referring to Figures S4-S7?"

**9.2 Author Response**

These now refer to Figures S9-S12.

**9.3 Manuscript Changes**

Lines removed.

---

## Author Comment (AC3)

**Point-by-Point Responses to Anonymous Referee #3**

We thank Anonymous Referee #3 for their time and effort in reviewing our work and appreciate their recommendations for ways to improve the manuscript. Following many of their suggestions, we believe the revised manuscript is an improvement upon the original submission. The referee comments below are taken from https://doi.org/10.5194/egusphere-2023-2227-RC3.

This document provides a point-by-point response to Referee #3. The responses follow the sequence: (1) comments from the referee, (2) authors' response, (3) authors' changes in manuscript.

**1.1 Referee Comment**

"Tribby and Wennberg argue in "An analysis coordinate transform to facilite [sic] use of in-situ aircraft observations for flux estimation" that simulated ethane and propane fields from the GEOS-Chem transport model run at three different horizontal resolutions look similar when the chemical tracer is plotted on a latitude versus potential temperature coordinate, although they look different when plotted on a latitude versus pressure coordinate. They then conduct a simple optimization in which the simulated mole fractions are compared to observed mole fractions from the ATom campaign, at the same latitude and potential temperature, and scaled to match. Tribby and Wennberg report that the scale factor is within 10% for simulations whose resolution ranges from 4x5 deg to 0.5x0.625 deg. They use this agreement to conclude that the use potential temperature as the analysis coordinate for comparing simulated to observed tracers could save time and money associated with chemical transport modeling."

**1.2 Author Response**

We have made substantial changes to the analysis and no longer present a Bayesian approach. We substantially revised the manuscript to focus on an analysis approach by creating pseudo data for ethane and propane as sampled by the DC8 aircraft during the Atmospheric Tomography (ATom) aircraft mission. We sample simulations from the highest resolution (0.5 x 0.625 degree) GEOS-Chem High Performance v14.3.1 and then evaluate the correlation of these pseudo data with lower resolution simulations and simulations sampled +/- 5 days from the pseudo data. We apply a simple statistical analysis that illustrates the value of potential temperature as a vertical coordinate in comparing sparse observations with a GCM.

**1.3 Manuscript Changes**

Figures 4-8.

**2.1 Referee Comment**

"The analysis in the paper was insufficient to substantiate this conclusion, and major revisions and additional writing are required before the paper is suitable for publication. There was no counterfactual presented, in terms of how close the optimization scale factors would be if a comparison were conducted in latitude/altitude space rather than latitude/potential temperature space. This comparison is critical to evaluate the authors' conclusion about the benefits of the proposed coordinate transform."

**2.2 Author Response**

See RC1 author response 6.2 for that addresses reasoning for change in approach. We now present a simple linear regression analysis, RMS, and RMSE that quantify the differences between the coordinate transform.

**2.3 Manuscript Changes**

Figure 4-8.

**3.1 Referee Comment**

"There was no argument offered for the appropriate spatial resolution at which fluxes could be optimized. While the methods were not clear, I got the impression that the authors were optimizing fluxes uniformly or perhaps zonally, both of which would be coarser than what is desired for many chemical species. The authors do not show whether (or how) this approach could be used to improve flux optimizations at continental, regional, or local scales, where fluxes would be optimized on a 2-d grid rather than a zonal a."

**3.2 Author Response**

As the Referee pointed out, this study focuses on constraining sources at large scale (coarse). In Tribby et al. 2022, using θ coordinate, we evaluated zonal errors in emission sources that were most clearly visible using summer data when the lifetimes of ethane and propane are shorter (Figure 3,4 in Tribby et al. 2022).

**3.3. Manuscript Changes**

No changes.

**4.1 Referee Comment**

"The spatial scales of the chemical transport model were also coarse compared to that which would be desired for some applications (e.g., 10s of km), and it is not clear whether this approach would add value at those scales."

**4.2 Author Response**

Even under those types of situations, traditional lat/lon/time/altitude approach can be problematic if sources are misattributed in space. (We are not trying to show this here.) Nevertheless, our new analysis approach shows that potential temperature can mitigate time and resolution errors when comparing coarse GCM simulations to sparse observations.

**4.3 Manuscript Changes**

Figures 4-8.

**5.1 Referee Comment**

"The authors simulate the chemical fields using fluxes that had previously been optimized in Tribby et al., 2022. Can this approach work when there are spatial biases in the distribution of prior fluxes fluxes (for example, zonal, meridional, or seasonal biases) imposed in the chemical transport model? If so, what are the limitations that are important for users of this approach to take into account?"

**5.2 Author Response**

The simulations have been previously optimized in Tribby et al. 2022 when there was a spatial bias in the distribution of underlying emissions. Using potential temperature as a dynamical coordinate supported revision of the underlying emissions grid. We added additional language detailing limitations for using this approach.

**5.3 Manuscript Changes**

Lines 53-60.

**6.1 Referee Comment**

"The authors describe the results plotted in Fig. 3 as being more consistent across resolutions than those plotted in Fig. 2, although the results for February 23 look quite different among the three resolutions even in Fig. 3. On this day, there are elevated concentrations in isolated patches within the middle of the free troposphere apparent in the 0.5x0.625 deg simulation, but not in the other simulations. The ATom observations are not shown for this day, so it is not clear whether the elevated concentrations were observed or whether they are an artifact of the the transport model. In either case, the paper should discuss what characteristics of the meteorology on this day lead to the result that the potential temperature approach does not lead to convergence among the various resolutions."

**6.2 Author Response**

While some days are predicted to have slightly higher mole fractions, these data did not appear to significantly affect or skew the new RMS, RMSE, or slope comparison analyses.

**6.3 Manuscript Changes**

Figure 4-8.

**7.1 Referee Comment**

"In general, the paper read as incomplete. It is noteworthy that the paper did not have a Discussion or Conclusions section. This should be remedied before a revision is considered. It ends abruptly and leaves the reader with many questions, and it is surprising that the paper went out for review. The methods section lacked sufficient detail to understand the simulations and emissions inventories underlying the simulations. I am not familiar with Tribby et al., (2022), and relevant aspects of the methodology, results, and conclusions should be summarized briefly so that readers of this paper are not required to do an in-depth read of that paper. The results section did not contain the relevant analysis and documentation to substantiate the conclusioins [sic] (as described above), and the lack of discussion meant that it is not clear what the caveats of employing this method were."

**7.2 Author Response**

See new Methods, Discussion and Conclusions section.

**7.3 Manuscript Changes**

Lines 52-251.